Economic value of trees in the estate of the Harewood House stately home in the United Kingdom

Peacock Julie 1
Ting Joey 2
Bacon Karen L. k.bacon@leeds.ac.uk 1
1 School of Geography, University of Leeds, University of Leeds , Leeds , United Kingdom
2 Department of Civil and Environmental Engineering and the School of Mining and Petroleum Engineering, University of Alberta , Edmonton , Canada
Mencuccini Maurizio
Electronic publication date: 2018 Sep 14
Publication date: 2018
Volume: 6
Electronic Location ID: e5411
Received 2018 Apr 6; Accepted 2018 Jul 30
Copyright: ©2018 Peacock et al.
Copyright year: 2018
Copyright holder: Peacock et al.
License: This is an open access article distributed under the terms of the Creative Commons Attribution License, which permits unrestricted use, distribution, reproduction and adaptation in any medium and for any purpose provided that it is properly attributed. For attribution, the original author(s), title, publication source (PeerJ) and either DOI or URL of the article must be cited.
License URL: https://creativecommons.org/licenses/by/4.0/

Keywords: Manor houses, Economic value, Trees, Natural capital, Ecosystem services, Stately homes

Funding: Queen Elizabeth II Diamond Jubilee Scholars Programme Joey Ting received funding from the Queen Elizabeth II Diamond Jubilee Scholars Programme. The funders had no role in study design, data collection and analysis, decision to publish, or preparation of the manuscript.

==============================
The estates of stately homes or manor houses are an untapped resource for assessing the ecosystem services provided by trees. Many of these estates have large collections of trees with clear value in terms of carbon storage, runoff prevention, and pollution removal along with additional benefits to biodiversity and human health. The estate of Harewood House in North Yorkshire represents an ideal example of such a stately home with a mixture of parkland and more formally planted gardens. The trees in each type of garden were analysed for height, diameter at breast height and light exposure. The data were then processed in iTrees software to generate economic benefits for each tree in both gardens. The analysis found that the larger North Front parkland garden had greater total benefits but the more densely planted formal West Garden had the greater per hectare value. In total, the trees on Harewood House estate are estimated to provide approximately £29 million in ecosystem service benefits. This study is the first to analyse the trees of stately homes for economic benefits and highlights that the trees are a valuable commodity for the estates. This should be considered in future planning and management of such estates.

Introduction

Ecosystem services refer to the services that ecological systems provide to humanity and numerous studies now attempt to put a financial value on such services in various ecosystems (Costanza et al., 1997; Costanza et al., 2014; De Groot et al., 2012). They can be considered from a local to a regional and even biome (Costanza et al., 1997; Bolund & Hunhammar, 1999) scale and are increasingly considered in conjunction with the development and implementation of policy (Egoh et al., 2007; De Groot et al., 2010; Wolff, Schulp & Verburg, 2015). The economic value and ecosystem service value of trees are increasingly well-studied in urban environments (e.g., Kowarik & Von der Lippe, 2018; Nowak et al., 2010 and references therein; Luttik, 2000), including large cities such as London, UK (London iTree Eco Project, 2015) and Chicago, USA (Nowak et al., 2010). These studies identified annual benefits derived from trees on the order of millions of pounds to the cities. For example, the trees of London were recently calculated to provide approximately £126.1 million in pollution removal value and £4.76 million per annum in carbon storage value (London iTree Eco Project, 2015).

Similar studies focusing on non-urban terrestrial ecosystem in agricultural (Tscharntke et al., 2005) and forest ecosystems (Gamfeldt et al., 2013; Ninan & Inoue, 2013) are also increasingly common and focus on a wide-range of ecosystem services such as human well-being (e.g. Fedele, Locatelli & Djoudi, 2017), biodiversity (e.g. Mace, Norris & Fitter, 2012) and carbon sequestration (e.g. Novara et al., 2017). One land-use type that does not currently feature in the ecosystem service literature is the estates of stately homes (sometimes also called manor homes). Stately homes are large houses that are or were occupied by landed gentry. In the United Kingdom, these houses and their often extensive grounds are now commonly open to the public as a means of generating income and may be located in rural areas or closer to city limits as cities have expanded since the houses were originally constructed. Many such grounds include a mixture of formally planted gardens, where both native and non-native trees and shrubs are grown for aesthetic purposes, and more open parkland environments. The gardens often preserve rare collections and varieties of different species, making them a useful conservation and research resource (Dosmann, 2006; Dosmann & Groover, 2012; Hockenberry Meyer et al., 2010). Additionally, their often extensive array of ecosystems mean that many estates are important sites for local biodiversity (Šantruckova et al., 2017; Ignatieva & Konechnaya, 2004; Lohumus & Liira, 2013). The grounds of stately homes, therefore, offer a currently untapped resource for considering the economic value of the trees in a managed ecosystem that often lies between rural and urban areas with a mixture of native and non-native species, similar to plant community composition along rural to urban gradients (Kowarik, 1990). Their positioning along the nexus of the urban–rural transition make them particularly interesting and offers a unique ecological placement for considering the value of the trees. With more and more of these properties meeting the costs of their upkeep by opening to the public and the expense of maintaining large parkland and gardens, the need to consider the ecosystem services and the potential economic value of their trees will become increasingly important. Considering the ecosystem services of stately homes and their estates has multiple potential benefits, including, raising the profile of the ecosystem service benefits of trees to the visiting public, providing a clear justification for the cost of maintaining the gardens; demonstrating what the gardens provide to the local environment and to the estates themselves; and showcasing the pollution-capture benefit of the trees (particularly for stately homes near busy roads and close to urban centres).

Harewood House, the location for this study, is located approximately 14 km from Leeds city centre in Yorkshire, United Kingdom. The house is over 260 years old and was constructed between 1756 and 1771. It sits on a 400 hectare estate that includes the Grade 1 listed main house building and publicly accessible facilities including a bird garden, café, and playground as well as parkland and more formal gardens. Harewood House is a member of the Treasure Houses of England and one of the foremost English stately homes. The diversity of ecosystems, different land use, and proximity to a large urban centre while still being primarily rural makes the Harewood House estate an ideal location to investigate the economic value of trees in these unique environments.

To the authors’ knowledge, this study provides the first such investigation of the economic benefits of the trees of stately homes. The aims of this study are (1) to determine the economic and ecosystem service value of trees in Harewood House and (2) to investigate if this differed between a parkland and a formal garden environment.

Materials & Methods

One parkland (the North Front) and one formal garden (the Western Garden) within the Harewood House estate were selected as representative of the two main types of land use within the estate for analysis. The North Front covers an area of 8.68 ha and the West Garden has an area of 0.98 ha. Both gardens were investigated to determine species composition and ecosystem service provision. All of the trees in both gardens (66 in the North Front and 57 in the West Garden) were identified to species level and measured for diameter at breast height (DBH), tree height, crown height, and light exposure measured by determining how many faces of the crown are exposed to light between 0 and 5. These data were then input to the iTree Eco programme (https://www.itreetools.org/) to calculate the economic value of each individual tree in terms of structural value, carbon storage, carbon sequestration, avoided runoff, pollution removal and energy savings using the methods outlined in both the iTree Eco programme and in the London iTree Eco Project (2015). In summary, the iTree Eco programme utilises measurements to calculate both total and annual economic values for the various outputs using reference data sets and standard equations based on the relationships of the simple plant measurements and the economic values. Measurements are separated by the programme into two sections, mandatory and recommended (iTree, 2017). Mandatory data are measurements that must be taken and inputted into the program for it to run, including DBH and species of tree. Recommended data are measurements that should be taken if possible, but are not required for the programme to run successfully. If the recommended data are not inputted, i-Tree Eco will estimate a value based on the DBH and species of the tree but the estimated value will not be as accurate as an actual measured value. Recommended data collected as part of this study were height of tree, height of canopy, size of canopy, light exposure, percent dieback, percent of crown missing, and land use type (iTree, 2017). We also recorded GPS locations for all trees, tree identity tag numbers, angle to bottom of canopy, height of canopy with respect to 0 degrees and height of canopy bottom (see Supplemental Information 1). Data collection was undertaken during July and August 2017.

All statistical data analysis was conducted using PAST version 3.15 and figures were prepared using SigmaPlot13. All data were tested for normality using the Anderson-Darling A test (all were non-normally distributed p < 0.05) and Mann Whitney U tests were used to test for difference to investigate non-normal data from the two gardens. The Supplemental Information 1 file provides all raw data used in the analyses.

Results

Composition of gardens

The North Front and the West Garden showed an array of differences in terms of their species composition and character. Table 1 summarises the diversity measures for both gardens and highlights that the West Garden has a higher richness and greater diversity than the North Front. There was also a difference in the density of trees between the two sites, with the North Front having 7.60 trees per hectare and the West Garden having 58.16 trees per hectare. There is a moderate turnover of species between the two gardens (Whittaker’s Beta W = 0.40) and the West Garden contains six of the 11 species in the North Front. Figure 1 shows the difference in relative abundance of each species planted in both gardens. The North Front has 11 species and is dominated by Quercus robur, which accounts for over 50% of the trees in this garden. The remaining 50% is split between several species at much lower abundance (<15% each), including Cedrus libani, Fagus sylvatica, Castanea sativa, Cedrus atlantica glauca, Cedrus atlantica, and Fagus sylvatica ‘Purpurea’ and other species at <2% abundance (one tree recorded in the garden). Species richness is greater in the West Garden (26 species), and there is a greater spread of relative abundance between species. The most common species remains Quercus robur but at a much reduced abundance (∼19% of trees in the garden) and with a larger array of species making up the rest of the garden’s trees, all at just over 12% or lower abundance. Rhododendron ponticum, Acer patmatum, Betuala pendula, and Cupressocyparis leylandii are the next most abundant species in the West Garden and none account for more than 12.2% of the trees in this garden.

Table 1 Diversity descriptive data comparing the North Front and West Garden.

Whittaker’s Beta W for the two gardens is 0.4, denoting a high turnover of species between the two gardens.

	North Front	West Garden	
Total individuals	66	57	
Species richness	11	26	
Shannon-Wiener H′	1.67	2.89	

Figure 1 Relative abundance (%) of trees in (A) the North Front parkland and (B) the West Garden.

Figure 2 Tree characteristics and differences between the North Front and West Garden: (A) Tree height (z =  − 2.13, P < 0.05); (B) DBH, (z =  − 3.66, P < 0.0005); (C) sides of canopy exposed to light (z =  − 4.11, P = 0.0001).

Tree characteristics

The trees in the North Front are significantly (p < 0.05) taller than those of the West Garden (Fig. 2A). DBH is also significantly greater (p < 0.005) in the North Front (Fig. 2B) with median height vales of trees in West Garden below the lower quartile of the trees in the North Front. The trees in the North Front have higher light exposure than those in the West Garden (Fig. 2C). There are three trees with only one face receiving light in the West Garden but none in the North Front receiving so little light. In addition, in the North Front there are 18 trees with all five sides exposed to light whereas there are just three trees that receive this level of light in the West Garden.

In addition to raw DBH analysis, size classes of DBH also differed considerably between the two gardens. Figure 3 shows the size class distribution of the trees in the North Front (black) and the West Garden (grey). The North Front has very few individual trees in the lower size classes with only five trees between 7 cm and 20 cm. The West Garden has more, with 15 trees in these small size classes. Both gardens have the majority of individuals in size classes between 60 cm and 100 cm.

Figure 3 Size class distribution based on diameter at breast height (DBH) of trees in the North Front (black) and West Garden (grey), and City of London (London iTree Eco Project, 2015) with London data provided by Treeconomics and Forest Research available at: http://www.urbantreecover.org.

Ecosystem services

Ecosystem service values were calculated for all measured trees in both gardens. The trees in the North Garden have a greater value, in terms of structure, carbon storage, and total annual benefits than the trees in the West Garden. However, per hectare, the West Garden’s trees have greater value. For the West Garden, nine trees with DBHs greater than 100 cm account for both 45% of the structural and carbon value and 31% of the total annual benefits. In the North Front trees with a DBH greater than 100 cm account for 32% and the value (structural and carbon storage) and 24% of the annual total benefits. However, if all trees with a DBH greater than 90 cm are included they account for 73% of the value (structural and carbon storage) and 67% of the annual total benefits. Detailed iTree output data are provided in Supplemental Information 1.

Discussion

Composition of Gardens

The cultural value of stately homes is well known and described (Cranz & Boland, 2004) with most stately homes highlighting their particular contribution to local cultural history in various manners through exhibitions and permanent displays. However, the ecological value, both in terms of biodiversity and ecosystem services (see below), are far less well studied (Lohumus & Liira, 2013). Both gardens provide habitat for various plants and animals and the higher biodiversity of the West Garden may be seen as a positive thing for wildlife. Quercus robur, the dominant species for both gardens has a high wildlife value, in terms of invertebrate populations, seeds to provide food, nesting space and fungal populations, although it has a low value to wildlife for pollen and nectar (Alexander, Butler & Green, 2006). This is similar to Fagus sylvatica and Castanea sativa, which both have high abundance in the North Front and are also found in the West Garden. Aesculus hippocastanum and Crataegus monogyna have individuals represented in the North Front with a high value to wildlife for pollen and nectar. In the west garden there are a wide range of species with a high wildlife value for both fruits and seeds and pollen and nectar including Sorbus aucuparia, Tilia platyphyllos, Ilex aquifolium, and Prunus pissardii (Alexander, Butler & Green, 2006). Other species in the West Garden have limited value for wildlife including Cupressocyparis leylandii and Rhododendron ponticum, although R. ponticum does attract and sustain pollinating insects (Stout, 2007). It is likely that the different species and different planting arrangements of the two gardens may support slightly different niches and increase overall biodiversity of the estate similar to the suggestions of Sjöman et al. (2016), but this was beyond the scope of the current study to investigate.

Ecosystem service value

This study identified a total of over £7 million for the structural and carbon storage value and over £1,000 of annual benefits for the two gardens within Harewood House. The formal West Garden had a higher value per unit area but the larger North Front parkland garden had a higher value overall. Both gardens together account for approximately 2.4% of the Harewood estate area and represent contrasting planting regimes—a lower density regime represented by the North Front and a higher density regime represented by the West Garden. Therefore, a conservative estimate of the total value provided by the trees to the Harewood House estate can be calculated by considering the average per area value of the two gardens and multiplying up to the entire estate area. This suggests that the total value to Harewood House of the trees is over £29 million for structural and carbon storage values and over £48K of annual benefits. The North Front represents a low tree density part of the estate and the West Garden represents a high-density planting part of the estate. Therefore, the average value of the two gardens per unit area represents a conservative estimate of the potential total value of trees to the estate.

Per hectare, the carbon storage values of the West Garden is far greater than the North Front. Harewood House estate is not a rural forest or plantation, nor is it a suburban or urban area. Therefore, it can be expected that the carbon storage values of the trees might sit somewhere between values previously reported for rural and urban environments. The North Front carbon storage value is below that reported as the average urban forest carbon storage density of 21.1 tC/ha and the average forest stand of 53.3 tC/ha in the USA (Nowak & Crane, 2002) at 16.942 tC/ha whereas the West Garden, at 80.047 tC/ha, exceeds both estimates. Dewar & Canell (1992) suggested that carbon storage in UK plantations ranged from approximately 40–80 tC/ha. This varied with species and thinning regime, but highlights that the North Front is below this average range in terms of carbon storage and the West Garden is at the very high end of the range for the UK.

Strohbach & Haase (2012) examined the range of carbon storage values across the city of Leipzig, Germany and found that the average carbon storage was 11.8 Mg C per hectare. The highest carbon storage was identified in areas of the city with 98.26 MgC/ha in areas of riparian forest and the lowest of ∼4 MgC/ha in areas with multi-story housing. Other studies have identified similar ranges for urban trees included Barcelona, Spain, at 11.2 average and 33.3 maximum (Chaparro & Terradas, 2009), Chicago, USA, at 14.1 average and 35.8 maximum (Nowak, 1994), Leicester, UK at 31.6 average and 288.6 maximum (Davis et al., 2011), and Hangzhou, China at 30.3 average (Zhao et al., 2010). This shows that the values calculated for Harewood House fall easily within the range of urban tree carbon storage and, for the West Garden, can be much higher than the average value for urban trees. The average values of Harewood House for carbon storage fall within or exceed these ranges, showing that the estates of stately homes can be a valuable carbon store, in addition to their ecological and cultural value. The trees in Harewood House also provide significant carbon sequestration, pollution removal and runoff reduction (Tables 2 and 3, Supplemental Information 1).

The distribution of trees into different size classes (Fig. 3) shows that Harewood House has a greater proportion of trees in larger (over 60 cm DBH) size classes than the smaller (20 cm or smaller) classes. In the North Front 68.1% of trees and 42.1% of trees in the West Garden have DBH of at least 60 cm. These trees are likely to be older and, usually, mature with some over 100 years old. When compared to the size class distribution of trees in London (see Fig. 6 in London iTree Eco Project, 2015) it is clear that the trees in Harewood House have a very different distribution across DBH size classes. Proportionally, there are far more trees in larger DBH size classes and far fewer smaller trees in Harewood House than in London. This suggests that Harewood House, and likely other similar stately homes with large estates, may provide a niche for the preservation of particularly large trees. Such trees have a significant and often disproportionate carbon-storage value (Diaz-Porras, Gaston & Evans, 2014) and are also likely to have additional ecological and cultural value. Contrastingly, the few, and in the case of the North Front very few, numbers of trees in the smaller DBH size classes may raise concerns for the long-term sustainability of the ecological and economic benefits of trees in the estate. This also suggests that the unique niche for larger, economically and ecologically valuable trees, may become reduced in the mid-term future. There is little regeneration of trees in either garden and the middle size classes are also quite sparse. This suggests that the planting of young trees should be prioritised while the larger trees are managed to ensure that they continue to provide carbon storage and other economic and ecological services for decades to come. These findings also highlight a clear management use for the iTree Eco programme and the use of similar approaches for future management of the estates of stately homes. This information could be useful in the future management of the estate by helping to inform decisions about which species to plant and if some trees do need to be removed, which species should be targeted and which avoided. A comparison to the full ecological value of the trees (e.g., provision of ecosystem and habitat for other organisms, impact on soil quality, among other services) is needed to supplement the economic information for a fully informed management system.

Table 2 Total benefits of the trees for the North Front and West Garden.

	Number of trees	Structural value	Carbon storage	Gross carbon sequestration	Avoided runoff	Pollution removal	Total annual benefits	
		(£)	(kg)	(£)	(kg/yr)	(£/yr)	(m3/yr)	(£/yr)	(kg/yr)	(£/yr)	(£/yr)	
North Front	66	447,918.08	147,058.82	9,411.79	3,005.00	192.35	154.63	124.78	75.05	496.70	816.55	
West Garden	57	238,989.56	78,446.80	5,020.56	1,553.00	99.41	61.57	49.96	30.01	199.53	348.86	

Table 3 Per hectare benefits of the trees in the North Front and West Garden.

	Number of trees	Structural value/ha	Carbon storage/ha	Gross carbon sequestration/ha	Avoided runoff/ha	Pollution removal/ha	Total annual benefits/ha	
		(£)	(kg)	(£)	(kg/yr)	(£/yr)	(m3/yr)	(£/yr)	(kg/yr)	(£/yr)	(£/yr)	
North Front	66	51,603.47	16,942.26	1,084.31	346.20	22.16	17.81	14.38	8.65	57.22	94.07	
West Garden	57	243,866.9	80,047.76	5,123.02	1,584.69	101.44	62.83	50.98	30.62	203.60	355.98	

Previous assessments of the natural value of stately homes has been largely descriptive (Halbrooks, 2005; Nestor & Mann, 1998) or semi-quantitative (Walerzak et al., 2015). Walerzak et al. (2015) highlight the necessity of being able to compare between studies and estates to fully understand the value and conservation requirements across a range of such properties. They propose a semi-quantitative approach that provides a numeric value for the value of the estate but do not fully consider the economic value of the natural capital within such estates. The iTree approach, while not considering the ecological value of the estates, provides a simple, reproducible and efficient method to generate economic values for the trees on these estates. The values presented in Tables 2 and 3 (and Supplemental Information 1) clearly show the economic value of the trees in the two gardens at Harewood House. They also show the difference between the two types of garden –the more densely planted West Garden had a greater per hectare value but the larger parkland garden of the North Front provides greater economic value overall. The findings suggest that a maximised economic value of planting could likely be achieved by increasing the density of planting in more areas of the estate; however, this would need to be balanced against the growth and maintenance of the larger trees (DBH > 60 cm) and the ecological and cultural values of the parkland. This approach provides clear data that can be compared easily between sites and, if used in conjunction with ecological data, could provide a valuable estimate of the value of the estates of stately homes.

Conclusions

The trees of the North Front and West Garden at Harewood House estate provide a range of important ecosystem services that amount to approximately £1,165 per year. This value is in addition to ecosystem provision and support of biodiversity on the estate and the well-documented health benefits of natural and parkland environments to people. This study shows that the economic value of trees in the estates of stately homes is not trivial and should be considered as part of the planning and management of such estates, and potentially, as part of the assessment of carbon budgets of local authorities and perhaps on larger scales across the country.

Supplemental Information

Supplemental Information 1 Raw data and iTree output for all measured trees

Click here for additional data file.

The authors would like to acknowledge Mr Trevor Nicholson, Head Gardener for the Harewood House Trust, for assistance with fieldwork and helpful discussion and Lord David Lascelles, 8th Earl of Harewood for permission to undertake fieldwork on the Harewood House estate. Data for DBH of London trees was provided by Treeconmics and Forest Research.

Additional Information and Declarations

Competing Interests

Author Contributions

Data Availability

The authors declare there are no competing interests.

Julie Peacock conceived and designed the experiments, analyzed the data, prepared figures and/or tables, authored or reviewed drafts of the paper, approved the final draft, conceived the original idea and designed the study.

Joey Ting performed the experiments, authored or reviewed drafts of the paper, approved the final draft, conducted fieldwork and refined study design.

Karen L Bacon analyzed the data, prepared figures and/or tables, authored or reviewed drafts of the paper, approved the final draft, led the drafting of the manuscript.

The following information was supplied regarding data availability:

Data is included in a Supplemental File.

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
