# Peer review of "Economic value of trees in the estate of the Harewood House stately home in the United Kingdom"

_PeerJ, doi:10.7717/peerj.5411_

## Round 0.1 · original submission · Major Revisions

Dear Author,

I am marking this editorial decision as Major Revision following one of the reviewers. I suggest you take the opportunity to thoroughly check the calculations and presentation of your data to avoid some of the mistakes spotted by one of the reviewers.

Reviewer 1 ·

Basic reporting

English used throughout the manuscript is clear, unambiguous and professional allowing easy and smooth reading. Introduction shows context and understanding of the topic and leads logically to the aims of this manuscript.
Literature used is relevant and covers the topic broadly. Precision of referencing is, however, poor. Already in the first section of the introduction, there are three mistakes: L 50 Hunhammer should be Hunhammar, L 54 Nowak is listed as Norwak and Luttik 2010 is listed as Luttik 2000 in the reference list. There are further referencing mistakes throughout the manuscript. The authors should thoroughly check the references.
Figures attached to the current file are of poor quality. Species names on Figure 1 are almost impossible to read. Caption of Figure 3 does not match to what is actually presented. Furthermore, there is a general problem with differences between the data and results presented in the text and on the figures/tables. For example, Table 3: Whittaker’s Beta W reported 0.33 in Table 1 and 0.4 on line 130; about 35% of relative abundance of Quercus robur on Figure 1, more than 50% on line 133 etc.
The general structure of the manuscript follows the PeerJ standards, with raw data supplied in a readable and understandable format.

Experimental design

The aims of the study are well defined, relevant and meaningful. It is stated how the current manuscript will fills an identified knowledge gap. Although the manuscript is focused on one single estate in United Kingdom, the results are discussed in much broader context, making it interesting for wide variety of readers.
Methods used in this paper could be described with bit more detail. For example, the measured light exposure of the trees is only explained in the result section on line 150. Although I understand that the iTrees software methods have been described elsewhere, it would be helpful to have some basic principles explained in the current manuscript as well. That would save the reader from having to scan through long reports or download user-manuals to get an idea of the methods used in the software.

Validity of the findings

Data collected for this manuscript is robust and statistically sound. However, discrepancies between the figures/tables and text raise some concerns. Results in the text do seem to match the raw data supplied.
The conclusions are linked to the original aims, limited to the results, yet discussed in a broader context

Additional comments

L 145 - 154: Something has happened there, as the same results are presented three times. Please edit this section.
Also section Ecosystem services (L 162 - 171) is bit difficult to read.

·

Basic reporting

The paper is written by professional English.
The paper as well as literature references deal mainly with Anglo-Saxon context. I suggest use more papers mainly from Germany (e.g. research of I. Kowarik and his team).
The paper has professional and clear structure.
Figures and tables are relevant.
Fig.1: The hight of columns does not correspond with information in text (lines 132 - 140).
Fig. 3: I suggest omit the column with London data that are not gained by authors´ own research. The caption of axis x should show the DBH interval in each columns.

Experimental design

The topic of the paper fits to scope of the journal.
The presented paper is a case study based on detailed enhancement of gardens in one estate. It is relevant for introduction of methodology how economic values of ecosystem services provided by gardens could be measured. Deep research of economic values of gardens should be based on more extent set of model areas.

Validity of the findings

Discussion and Conclusion are well stated and linked with research questions. However, the estimation of the value provided by trees to the estate (lines 199 - 202) should be more specified. It could be supposed that not only number of trees but also their composition (species richness, health conditions) will differ (be higher) in gardens than in common agrucultural and/or periurban landscape. How the presented estimation was made?

Additional comments

The paper is based on one case study, nevetheless, it brings important findings and is worth to be published.

---

## Round 0.2 · accepted · Accept

Dear authors,
I am happy with the changes implemented following the first round of reviews.

Both reviews have highlighted the significance of the paper for current discussions on this theme. One of the reviewers recommended major changes, however having reviewed your manuscript, I am happy to recommend acceptance based on your current submission, as most of the requested changes were of a typographical nature. I am now happy that the quality of both your tables and figures has now improved and brought to publication level.

#